# Acute kidney injury in critically ill cancer patients is associated with mortality: A retrospective analysis

Nina Seylanova[1,2], Siobhan Crichton[3], Jing Zhang[1,4], Richard Fisher[5], Marlies Ostermann[1] *

**1** Department of Critical Care, King's College London, Guy's & St Thomas' Hospital Foundation Trust, London, United Kingdom, **2** Sechenov Biomedical Science and Technology Park, Sechenov First Moscow State Medical University, Moscow, Russian Federation, **3** Medical Research Council Clinical Trials Unit, University College London, London, United Kingdom, **4** Department of Critical Care Medicine, Zhongnan Hospital of Wuhan University, Wuhan, China, **5** Department of Critical Care, King's College Hospital, London, United Kingdom

* Marlies.Ostermann@gstt.nhs.uk

## Abstract

### Background

In critically ill patients, acute kidney injury (AKI) is common and associated with short- and long-term complications. Our objectives were to describe the epidemiology and impact of AKI in cancer patients admitted to the Intensive Care Unit (ICU).

### Methods

We identified all patients with a haematological malignancy (HM) or solid tumour (ST) who had an emergency admission to the ICU in a tertiary care centre between January 2004 and July 2012. AKI was defined according to the KDIGO criteria.

### Results

429 patients were included of whom 259 (60%) had AKI. Among HM patients, 73 (78%) had AKI (70% AKI on admission to ICU; 7% during ICU stay); among ST patients, 186 (56%) had AKI (45% on admission to ICU, 11% during ICU stay). ICU and 28-day mortality rates were 33% and 48%, respectively in HM patients, and 22% and 31%, respectively in ST patients. Multivariable analysis showed that AKI was an independent risk factor for both ICU and 28-day mortality. New AKI after 24 hours in ICU was associated with higher mortality than AKI on admission.

### Conclusions

AKI is common in critically ill cancer patients and independently associated with ICU and 28-day mortality.

**Data Availability Statement:** All relevant data are within the manuscript and its Supporting Information files.

**Funding:** The author(s) received no specific funding for this work.

**Competing interests:** The authors have declared that no competing interests exist.

## Introduction

The number of cancer patients requiring admission to the Intensive Care Unit (ICU) has gradually increased in the last 2 decades [1,2]. Data from a large international observational study showed that 1 in 7 patients admitted to general ICUs in Europe had cancer [3]. Although the prognosis of cancer patients in general has improved, acute kidney injury (AKI) is a frequent and serious complication [4–8]. Analysis of the Danish Cancer Registry including >37.000 patients with cancer revealed that the 1-year and 5-year risk of AKI were 17.5% and 27.0%, respectively [4]. Furthermore, 5.1% of patients in whom AKI developed required long-term dialysis within 1 year.

The risk of AKI is highest during critical illness ranging from 54% in critically ill unselected cancer patients, to up to 69% in patients with a haematological malignancy [5,7,9,10]. However, reports vary depending on the criteria used to define AKI. To our best knowledge, only two studies used the current KDIGO classification to define AKI in cancer patients [5,6]. In addition, the types of patients included vary between studies: some investigators analysed all cancer patients, including those admitted to the ICU following elective surgery, whereas others focussed on specific subgroups, for instance, patients who required emergency admission to ICUs, patients receiving chemotherapy in the ICU or patients on mechanical ventilation [8].

Our objectives were a) to describe the epidemiology of AKI in cancer patients with an unplanned emergency admission to the ICU using the most recent consensus criteria for AKI; b) to differentiate between AKI on admission and new AKI acquired during ICU, and c) to explore the impact of AKI on risk of mortality.

## Materials and methods

### Setting

Guy's & St Thomas' NHS Foundation Trust is a tertiary referral centre for haematology and oncology. Oncology related in- and out-patient services are based at the Guy's Hospital site where critical care support is provided in a 13-bed Consultant-led multidisciplinary ICU. At Guy's Hospital, there is no Emergency Department, and referrals to the ICU are made predominantly by inpatient teams caring for patients on the medical and surgical wards and from the chemotherapy day-unit, as well as from other hospitals within the region. It is departmental policy that all admissions to the ICU are approved by the ICU consultant in charge. Decisions regarding the care of cancer patients are made by the ICU consultant in collaboration with the consultant-led oncology and haematology team who are available on a 24-hour basis. Full multiorgan support including haemodynamic, renal and advanced respiratory support can be provided at all times. The ICU has a fully computerised electronic patient record system where all therapies and laboratory data are recorded at time of generation.

### Study design

We retrospectively screened the records of all admissions to the ICU between January 2004 and July 2012 and identified adult patients (18 years or older) with a diagnosis of malignancy admitted as an emergency. Patients with planned admissions following elective surgery and those who spent <24 hours in ICU were not included. In case of multiple admissions, only the first one was considered.

### Data collection

We screened the patients' computerised electronic medical notes and laboratory records and collected the following data: demographics, type of malignancy, site of primary tumour,

presence of metastases in patients with solid tumours, history of bone marrow transplantation (BMTx) in patients with haematological malignancies, number of ICU admissions, presence of sepsis, neutropenia (defined as white cell count <1.0×109/L), thrombocytopenia (defined as platelet count <20×109/L), Acute Physiology and Chronic Health Evaluation (APACHE) II [11] and Sequential Organ Failure Assessment (SOFA) Score [12,13] on admission to ICU.

Ventilatory support was defined as the need for mechanical ventilation via an endotracheal tube or non-invasive ventilation via a face-mask; cardiovascular support was defined as the need for a continuous infusion of an inotropic or vasopressor drug. Renal replacement therapy (RRT) included continuous or intermittent modalities. Decisions to initiate organ support were individualised and made by the ICU team on a patient by patient basis.

AKI was defined according to the serum creatinine criteria of the Kidney Disease: Improving Global Outcomes (KDIGO) classification [14]. The lowest serum creatinine concentration within 3 months before ICU admission was used as baseline value. For patients without any previous serum creatinine results available, and with no past history of renal disease, the baseline creatinine was back-calculated using the Modification of Diet in Renal Disease formula (four-variable equation), assuming an estimated glomerular filtration rate of 75 mL/min/1.73 m2 before ICU admission [14]. Maximum stage of AKI was recorded based on the maximum change in serum creatinine from baseline. We differentiated between AKI on admission to ICU (i.e. within 24 hours of ICU admission) versus AKI that developed after 24 hours in ICU. Severe AKI was defined as AKI stage 2 or 3. Patients with end-stage renal failure or a kidney transplant were excluded. The main outcomes were ICU and 28-day mortality.

## Statistical analysis

Categorical data were summarised as frequency (percentage), and continuous variables were summarised as median [interquartile range (IQR)]. Univariable comparisons of the characteristics of patients who left the ICU alive and survived to 28 days and non-survivors were made using a χ2, Fisher's exact or Mann-Whitney test, as appropriate.

Multivariable logistic regression models were used to explore the association between mortality and AKI (no AKI, AKI on admission or AKI developed during stay), with forward stepwise procedure (with variables entered into the model using a threshold of p<0.1) used to adjust for and identify factors known at the time of admission which were independently associated with ICU and 28-days mortality. To avoid issues of multicollinearity, the renal component was excluded from the SOFA Score in the models. APACHE II was excluded for the same reason. A p-value of 0.05 was considered statistically significant. Analysis was carried out using IBM SPSS v23 and STATA/IC 14.

## Ethics

The study had institutional approval. Formal review by a Research Ethics Committee and need for individual informed consent were not required since the research was limited to secondary use of data previously collected in the course of normal care and the patients were not identifiable to the research team carrying out the research (Governance Arrangements for Research Ethics Committees published by the UK Health Departments).

## Results

### Demographics and patient characteristics

During the 8-year study period from January 2004 until July 2012, 473 cancer patients had an unplanned emergency admission to the ICU at Guy's Hospital. Forty-four patients met one or

more exclusion criteria. The remaining 429 patients were included in the analysis. Their ICU mortality was 24.5% and 28-day mortality was 34.7%. Multi-organ failure was the cause of death in 90% of patients who died in the ICU.

## Haematological malignancies

Ninety-four patients (22%) had a haematological malignancy (median age 57 [IQR 42–64]; 66% male). (Table 1) The most common types were lymphoma (44%), leukaemia (43%) and myeloma (14%); 35% of patients had had a bone marrow transplant. 32% of patients had more than one unplanned admission to the ICU during the same hospitalisation. 42 (45%) and 48 (51%) patients required cardiovascular and ventilatory support during their stay in ICU, respectively. On admission to ICU, the median SOFA score was 8 [IQR 6–10], the median APACHE II score was 20 [IQR 16–24], and 82% of patients had sepsis. ICU and 28-day mortality were 33% and 48%, respectively. 62% of patients did not survive the following 180 days, and 68% died during the first year from admission. (Table 1) 73 (78%) patients met criteria for AKI. The majority (90%) had AKI on admission to ICU and 26% were treated with RRT.

## Solid tumours

335 patients (78%) had a solid tumour of whom 106 (32%) were known to have metastatic disease at time of ICU admission. (Table 1) Their median age was 65 years [IQR 56–73] and 60% were male. The most common types of tumours were lung (38%), head/neck (16%), genitourinary (16%), oesophageal (7%), breast (5%) and colorectal cancer (5%). On admission to ICU, the median SOFA and APACHE II Scores were 4 [IQR 3–7] and 18 [IQR 14–21], respectively, and 69% of patients had sepsis. 37% of patients needed cardiovascular support and 61% of patients had ventilatory support. 78% of patients with solid tumours were discharged alive from the ICU and 69% survived to day 28. 180 and 360-day mortality rates were 54% and 62%, respectively. 186 (56%) patients met criteria for AKI of whom 80.6% had AKI on admission to ICU and 36 developed AKI whilst in ICU. (Table 1) AKI stage 1 was the most common form of AKI. Forty-eight patients (14%) received RRT.

## Renal replacement therapy

Seventy-two patients received RRT of whom 43 (59.7%) were started on continuous renal replacement therapy (CRRT) within 24 hours of admission. Their ICU mortality was 44.2%. The remaining 29 patients were initiated on CRRT at a median time of 4 days. Their ICU mortality was 72.4% ($p < 0.05$).

## Risk factors for mortality

In univariable analyses, cancer patients who died in ICU were characterised by a significantly higher SOFA and APACHE II score on admission to ICU, a significantly higher rate of sepsis, thrombocytopenia, neutropenia, and AKI and a higher proportion requiring cardiovascular and/or ventilatory support. (Table 2) These factors were also significantly different between 28-day survivors and non-survivors apart from sepsis. (Table 2) There was no significant difference in age and gender.

## Multivariable analysis

In multivariable logistic regression models, AKI and higher SOFA Score were independently associated with ICU and 28-day mortality. (Table 3) Development of new AKI during stay in

**Table 1. Demographics and outcomes.**

| Parameter | Patients with haematological malignancy (n = 94) | Patients with solid tumour (n = 335) |
|---|---|---|
| **Age** (years), median (IQR) | 57 (42–64) | 65 (56–73) |
| **Male gender**, n (%) | 62 (66) | 201 (60) |
| **Number of ICU admissions during same hospital admission** | | |
| 1 admission, n (%) | 64 (68.1) | 286 (85.4) |
| 2 admissions, n (%) | 19 (20.2) | 37 (11) |
| ≥3 admissions, n (%) | 11 (11.7) | 12 (3.6) |
| **Type of malignancy** | | |
| Myeloma, n (%) | 13 (13.8) | |
| Leukaemia, n (%) | 40 (42.6) | |
| Lymphoma, n (%) | 41 (43.6) | |
| BMTx, n (%) | 33 (35.1) | |
| Lung cancer, n (%) | | 128 (38.2) |
| Head & Neck cancer, n (%) | | 54 (16.1) |
| Genitourinary cancer, n (%) | | 52 (15.5) |
| Oesophageal cancer, n (%) | | 23 (6.9) |
| Breast cancer, n (%) | | 16 (4.8) |
| Colorectal cancer, n (%) | | 15 (4.5) |
| Other*, n (%) | | 47 (14) |
| Known metastatic disease, n (%) | | 106 (31.6) |
| **Severity of illness on admission to ICU** | | |
| SOFA score, median (IQR) | 8 (6–10) | 4 (3–7) |
| APACHE II score, median (IQR) | 20 (16–24) | 18 (14–21) |
| Sepsis, n (%) | 77 (81.9) | 232 (69.3) |
| Thrombocytopenia, n (%) | 35 (37.2) | 9 (2.7) |
| Neutropenia, n (%) | 52 (55.3) | 33 (9.9) |
| Need for inotropes, n (%) | 36 (38.3) | 73 (21.8) |
| Need for mechanical ventilation, n (%) | 45 (47.9) | 156 (46.6) |
| **AKI on admission**, n (%) | 66 (70.2) | 150 (44.8) |
| AKI stage 1, n (%) | 26 (27.7) | 61 (18.2) |
| AKI stage 2, n (%) | 14 (14.9) | 35 (10.4) |
| AKI stage 3, n (%) | 26 (27.7) | 54 (16.1) |
| **AKI during ICU stay**, n (%) | 7 (7.4) | 36 (10.7) |
| AKI stage 1, n (%) | 1 (1.1) | 19 (5.7) |
| AKI stage 2, n (%) | 2 (2.1) | 5 (1.5) |
| AKI stage 3, n (%) | 4 (4.3) | 12 (3.6) |
| RRT on admission, n (%) | 16 (17) | 27 (8.1) |
| RRT at any time during stay, n (%) | 23 (24.5) | 49 (14.6%) |
| **Outcomes** | | |
| ICU length of stay (days), median (IQR) | 4 (3–8) | 4 (2–7) |
| ICU mortality, n (%) | 31 (33) | 74 (22.1) |
| 28-day mortality, n (%) | 45 (47.9) | 104 (31) |
| 180-day mortality, n (%) | 58 (61.7) | 180 (53.7) |
| 360-day mortality, n (%) | 64 (68.1) | 208 (62.1) |

Abbreviations: AKI = acute kidney injury; APACHE = Acute Physiology and Chronic Health Evaluation; BMTx = bone marrow transplantation; ICU = intensive care unit; IQR = interquartile range; RRT = renal replacement therapy; SOFA = Sequential Organ Failure Assessment

* mesothelioma (n = 8), ovarian (n = 8), cervical (n = 5), brain (n = 7), endometrial (n = 3), pancreas (n = 3), vascular (n = 2), neuroendocrine (n = 2), uncertain gastrointestinal stromal tumour (n = 2), germ cell (n = 2), biliary tract (n = 1), thyroid (n = 1), thymoma (n = 1), carcinoid (n = 1), undifferentiated malignancy (n = 1).

**Table 2. Unadjusted comparison between survivors and non-survivors.**

| Parameter | Cancer Patients (solid tumour or haematological malignancy) (n = 429) | | | | | |
|---|---|---|---|---|---|---|
| | ICU survivors (n = 324) | ICU non-survivors (n = 105) | p—value | 28-day survivors (n = 278) | 28-day non-survi-vors (n = 149) | p—value |
| Age (years), median (IQR) | 64 (55–71) | 62 (52–72) | 0.385 | 64 (55–71) | 62 (52–71) | 0.319 |
| Male gender, n (%) | 201 (62) | 62 (59) | 0.585 | 173 (62.2) | 89 (59.7) | 0.613 |
| ICU LOS (days), median (IQR) | 4 (2–7) | 5 (3–11) | 0.015 | 4 (2–8) | 4 (2–7) | 0.949 |
| Haematological malignancy, n (%) | 63 (19.4) | 31 (29.5) | 0.030 | 49 (17.6) | 45 (30.2) | 0.003 |
| Severity of illness at ICU admission | | | | | | |
| SOFA score, median (IQR) | 5 (3–7) | 8 (5–11) | <0.001 | 4 (3–7) | 7 (4–9) | <0.001 |
| SOFA score minus Renal, median (IQR) | 4 (2–6) | 7 (4–9) | <0.001 | 4 (2–6) | 6 (3.5–9) | <0.001 |
| APACHE II score, median (IQR) | 17 (13–21) | 21 (18–26) | <0.001 | 17 (13–20) | 20 (16–26) | <0.001 |
| Sepsis at admission, n (%) | 226 (69.8) | 83 (79) | 0.046 | 195 (70.1) | 112 (75.2) | 0.226 |
| Thrombocytopenia, n (%) | 25 (7.7) | 19 (18.1) | 0.002 | 21 (7.6) | 23 (15.4) | 0.011 |
| Neutropenia, n (%) | 51 (15.7) | 34 (32.4) | <0.001 | 40 (14.4) | 45 (30.2) | <0.001 |
| AKI at any time, n (%) | 170 (52.5) | 90 (85.7) | <0.001 | 146 (52.5) | 113 (75.8) | <0.001 |
| No AKI, n (%) | 155 (47.8) | 15 (14.3) | <0.001 | 133 (47.8) | 36 (24.2) | <0.001 |
| AKI on admission, n (%) | 149 (46) | 67 (63.8) | | 125 (45) | 91 (61.1) | |
| AKI during stay, n (%) | 20 (6.2) | 23 (21.9) | | 20 (7.2) | 22 (14.8) | |
| No AKI 2/3, n (%) | 232 (71.6) | 45 (42.9) | <0.001 | 198 (71.2) | 77 (51.7) | <0.001 |
| AKI 2/3 on admission, n (%) | 86 (26.5) | 43 (41) | | 73 (26.3) | 56 (37.6) | |
| AKI 2/3 during stay, n (%) | 6 (1.9) | 17 (16.2) | | 7 (2.5) | 16 (10.7) | |
| RRT at any time, n (%) | 32 (9.9) | 40 (38.1) | <0.001 | 30 (10.8) | 42 (28.2) | <0.001 |
| Cardiovascular support on admission to ICU, n (%) | 59 (18.2) | 50 (47.6) | <0.001 | 52 (18.7) | 57 (38.3) | <0.001 |
| Ventilatory support on admission to ICU, n (%) | 134 (41.4) | 67 (63.8) | <0.001 | 108 (38.8) | 91 (61.1) | <0.001 |

Abbreviations: AKI = acute kidney injury; APACHE = Acute Physiology and Chronic Health Evaluation; BMTx = bone marrow transplantation; ICU = intensive care unit; IQR = interquartile range; LOS = length of stay; RRT = renal replacement therapy; SOFA = Sequential Organ Failure Assessment

**Table 3. Multivariable analysis: Factors affecting ICU and 28-day survival.**

| Parameter | ICU mortality | | | 28-day mortality | | |
|---|---|---|---|---|---|---|
| | OR | 95% CI | p—value | OR | 95% CI | p–value |
| **No AKI at any time in ICU** | 1 | | | 1 | | |
| **AKI on admission** | 4.05 | 2.04–8.05 | | 2.24 | 1.35–1.70 | |
| **AKI developed during ICU stay** | 10.72 | 4.43–25.96 | **<0.001** | 2.86 | 1.32–6.16 | **0.0025** |
| **SOFA Score minus Renal on admission to ICU** | 1.27 | 1.16–1.38 | **<0.001** | 1.23 | 1.14–1.33 | **<0.001** |
| **Age** | 1.00 | 0.98–1.02 | 0.88 | 0.999 | 0.98–1.02 | 0.94 |
| **Sepsis on admission to ICU** | 0.97 | 0.51–1.87 | 0.93 | 0.92 | 0.54–1.56 | 0.75 |
| **Cancer type** | 0.89 | 0.46–1.72 | 0.77 | 1.05 | 0.59–1.86 | 0.88 |
| **Neutropenia on admission to ICU** | 1.17 | 0.59–2.31 | 0.66 | 1.18 | 0.63–2.22 | 0.60 |
| **Male gender** | 0.81 | 0.48–1.39 | 0.45 | 0.83 | 0.524–1.32 | 0.43 |

Abbreviations: AKI = acute kidney injury; APACHE = Acute Physiology and Chronic Health Evaluation; CI = confidence interval; ICU = intensive care unit; IQR = interquartile range; OR = odds ratio; SOFA = Sequential Organ Failure Assessment; Variables included in the final multivariable model were selected using a forward selection procedure. ORs for variables that were not included were estimated by adding each of the variables to the multivariable model in turn.

**Table 4. Multivariable analysis: Factors affecting 28 day survival.**

| Parameter | Patients with solid tumours n = 335 | | | Patients with haematological malignancies, n = 94 | | |
|---|---|---|---|---|---|---|
| | OR | 95% CI | p—value | OR | 95% CI | p–value |
| **No AKI at any time in ICU** | 1 | | | 1 | | |
| **AKI on admission** | 1.74 | 0.98–3.10 | | 9.69 | 1.75–53.62 | |
| **AKI developed during ICU stay** | 3.07 | 1.32–7.17 | **0.022** | 7.52 | 0.58–96.98 | **0.034** |
| **SOFA Score minus Renal on admission to ICU** | 1.25 | 1.14–1.38 | **<0.001** | 1.49 | 1.15–1.92 | **0.002** |
| **Age** | 0.997 | 0.98–1.02 | 0.76 | 1.05 | 1.00–1.10 | **0.033** |
| **Sepsis on admission to ICU** | 1.29 | 0.71–2.34 | 0.41 | 0.13 | 0.02–0.77 | **0.025** |
| **Neutropenia on admission to ICU** | 1.99 | 0.78–5.08 | 0.15 | 0.36 | 0.89–1.45 | 0.15 |
| **Male gender** | 1.22 | 0.71–2.12 | 0.47 | 0.43 | 0.13–1.41 | 0.16 |
| **Known metastatic disease** | 3.44 | 1.98–5.98 | **<0.001** | - | - | - |
| **Bone marrow transplant** | - | - | - | 0.62 | 0.18–2.12 | 0.44 |

Abbreviations: AKI = acute kidney injury; CI = confidence interval; ICU = intensive care unit; OR = odds ratio; SOFA = Sequential Organ Failure Assessment

the ICU was independently associated with a higher risk of mortality than AKI that was present on admission to ICU.

We repeated the multivariable analyses for patients with haematological malignancies and solid tumours separately. (Table 4)

In patients with a haematological malignancy, age, AKI and severity of non-renal illness on admission to ICU were independent risk factors for 28-day mortality. Among patients with a solid tumour, AKI, severity of non-renal illness on admission to ICU and known metastatic disease were independently associated with 28-day mortality.

## Discussion

The key findings of this study are that AKI is very common in cancer patients requiring emergency admission to the ICU. Second, most AKI is already present on admission to ICU rather than acquired in ICU. Third, AKI in cancer patients is independently associated with mortality, and forth, new AKI during stay in the ICU is independently associated with a higher risk of ICU and 28-day mortality than AKI that is present on admission.

The association between cancer and kidney disease has long been known. This was formally acknowledged by the creation of onco-nephrology as a subspecialty of nephrology [15,16]. Cancer patients are particularly vulnerable to AKI. The exact incidence and severity vary depending on the type and stage of cancer, the treatment regimen and co-existing acute and chronic co-morbidities [7].

Using most recent consensus criteria for AKI, our data confirm that AKI is indeed a serious complication in cancer patients who require emergency admission to the ICU. A recent Systematic Review with Meta-Analysis of individual data including 7,354 patients confirmed that the mortality of critically ill cancer patients had decreased during the last decades and that there was a significant reduction in mortality in cancer patients requiring organ support in the ICU, except for patients requiring RRT [17].

The exact reasons for the high mortality in cancer patients with AKI are not known but several potential explanations exist. AKI increases the risk of toxic effects from systemic chemotherapy and jeopardizes the continuation of cancer therapy. It also limits patient participation in possibly life-saving clinical trials. In patients treated with potentially curative regimens, AKI makes drug dosing challenging and may necessitate dose reductions or the use of alternative regimens with better renal safety records but less efficacy.

Using the most recent consensus criteria to define AKI, our study is the largest to date confirming the serious prognostic impact of AKI in cancer patients requiring emergency admission to the ICU. More work is necessary to explore the underlying causes and importantly to identify potentially reversible factors.

Despite this strength, it is important to acknowledge some potential limitations. Our analysis has all limitations of a retrospective single centre study with a heterogeneous patient population. In particular, our analysis was limited to data that were routinely collected in all patients. As a result, we do not have any information on other important aspects, for instance health care costs, or patient centered outcomes like quality of life. Nevertheless, our study included a large cohort of patients with common types of malignancies typically admitted to ICUs in hospitals with large oncology and haematology units. Second, we defined AKI by the KDIGO definition but only used the creatinine criteria. The main reason was that hourly urine output criteria were not available for all patients, in particular patients with neutropenia where insertion of a urinary catheter was avoided if possible. As such, it is possible that the prevalence of AKI was underestimated. We also acknowledge that a proportion of patients had low muscle mass and/or sepsis which may affect creatinine production. As such, it is possible that the overall incidence of AKI was underestimated and that the correlation between AKI and mortality was overestimated. Third, we were unable to identify the exact causes of AKI. The majority of patients had sepsis and multi-organ dysfunction on admission to the ICU which implies that AKI was most likely multi-factorial. No patient had a renal biopsy. Fourth, we were not able to explore the exact causes of death but confirm that 90% of patients died from multi-organ failure. Finally, we acknowledge that the current consensus criteria of AKI have limitations and that we may have underestimated the incidence of AKI, especially in patients with decreased creatinine production secondary to loss of cell mass, low protein intake and associated liver dysfunction. Novel AKI biomarkers are available but not routinely used in clinical practice.

## Conclusions

AKI is a serious complication in critically ill cancer patients and independently affects the chances of survival. Our data suggest that more work is necessary.

## Supporting information

**S1 Data.**
(XLSX)

## Author Contributions

**Conceptualization:** Nina Seylanova, Marlies Ostermann.

**Formal analysis:** Nina Seylanova, Siobhan Crichton.

**Investigation:** Nina Seylanova.

**Methodology:** Nina Seylanova, Siobhan Crichton, Jing Zhang, Richard Fisher, Marlies Ostermann.

**Supervision:** Marlies Ostermann.

**Writing – original draft:** Nina Seylanova.

**Writing – review & editing:** Siobhan Crichton, Jing Zhang, Richard Fisher, Marlies Ostermann.

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
