## [Decision Letter · Decision Letter 0]

2 Mar 2020

PONE-D-20-01764

Acute kidney injury in critically ill cancer patients is associated with mortality: a retrospective analysis

PLOS ONE

Dear Dr. Ostermann,

Thank you for submitting your manuscript to PLOS ONE. After careful consideration, we feel that it has merit but does not fully meet PLOS ONE’s publication criteria as it currently stands. Therefore, we invite you to submit a revised version of the manuscript that addresses the minor points raised during the review process by reviewer 1.

We would appreciate receiving your revised manuscript by Apr 16 2020 11:59PM. To enhance the reproducibility of your results, we recommend that if applicable you deposit your laboratory protocols in protocols.io, where a protocol can be assigned its own identifier (DOI) such that it can be cited independently in the future. For instructions see: http://journals.plos.org/plosone/s/submission-guidelines#loc-laboratory-protocols

We look forward to receiving your revised manuscript.

Kind regards,

Emmanuel A Burdmann

Academic Editor

PLOS ONE

Journal Requirements:

2. We noticed you have some minor occurrence(s) of overlapping text with the following previous publication(s), which needs to be addressed:

http://dx.doi.org/10.1136/bmjopen-2016-011363

In your revision ensure you cite all your sources (including your own works), and quote or rephrase any duplicated text outside the Methods section. Further consideration is dependent on these concerns being addressed.

Reviewers' comments:

Reviewer's Responses to Questions

**Comments to the Author**

1. Is the manuscript technically sound, and do the data support the conclusions?

Reviewer #1: Yes

Reviewer #2: Yes

2. Has the statistical analysis been performed appropriately and rigorously? 

Reviewer #1: Yes

Reviewer #2: Yes

3. Have the authors made all data underlying the findings in their manuscript fully available?

Reviewer #1: Yes

Reviewer #2: Yes

4. Is the manuscript presented in an intelligible fashion and written in standard English?

Reviewer #1: Yes

Reviewer #2: Yes

5. Review Comments to the Author

Reviewer #1: I would like to congratulate the authors on a very well written manuscript. It is well known in the literature that Acute Kidney Injury is prevalent in 5-20% of patients admitted to the ICU and this is generalized, not specific to cancer patients. This is a very difficult subset of patients to analyze in order to determine cause of mortality as multiple cofounding factors exist.

The authors did a nice job of addressing the main limitations with the study but have a few comments/questions that the authors might address in the Discussion section.

1. The use of SCr only as a marker of renal function. The authors explained why they did not want to place foley catheters in many patients but as many cancer patients have cachexia and very small muscle mass which may affect SCr, knowing that the overall incidence of AKI could have been underestimated the correlation between AKI and mortality may have been overestimated. Could the authors comment on this.

2. The inability to determine exact cause of death was also an issue as the majortiy of patients with AKI had Sepsis. Sepsis is known to be the most common cause of AKI in the ICU and often is a reversible cause of AKI. Were the authors able to determine if the sepsis was fully treated prior to death.

3. There is significant data to support the use of early RRT once AKI is diagnosed. The authors mentioned both Intermittent modes as well as Continuous modes were utilized. Did they notice a difference in timing of initiation or mode of dialysis with respect to cause of or timing of mortality.

Reviewer #2: The authors studied the impact of AKI in cancer patients admitted to an ICU and 28 day survival and overall survival. It has included both haematological malignancy (HM) and solid tumour (ST) for a total of 429 patients of which 60% has experienced AKI in total. Multivariable analysis showed that SOFA Score minus Renal

on admission to ICU and AKI was an independent risk factor for both ICU and 28-day mortality. Interestingly, new AKI after 24 hours in ICU was associated with higher mortality than AKI on admission.

comments: It has been demonstrated that AKI in cancer patients is associated with increased mortality. I agree this is one of few papers looking in to ICU admitted patients. The non-survivors overall and 28 days non-survivors had a higher median SOFA scores and AKI percentage. Interestingly the percentage of patients on RRT was higher in the non-survivors in both categories.

In the univariate analysis there is no comment about the use of RRT in this population and if it had a negative impact on survival would be interesting.

Initially, the authors do separate the hematological malignancies from solid tumor but in the analysis place them together. Typically patients with hematological malignancies are typically sicker and have poorer mortality in the ICU especially stem cell transplant patients; therefore, placing them all in the same category for survival analysis maybe inaccurate. I do agree that separating the in to hematological vs solid malignancy will limit numbers etc for the analysis.

Looking in to new AKI in the ICU is not well studied and this finding further contributes to the literature.

6. PLOS authors have the option to publish the peer review history of their article (what does this mean?). If published, this will include your full peer review and any attached files.

Reviewer #1: No

Reviewer #2: No

---

## [Author Response · Author response to Decision Letter 0]

22 Mar 2020

Responses to reviewers

We have checked the instructions again and revised the paper according to the style requirements of the journal. 

2. We noticed you have some minor occurrence(s) of overlapping text with the following previous publication(s), which needs to be addressed:

http://dx.doi.org/10.1136/bmjopen-2016-011363 In your revision ensure you cite all your sources (including your own works), and quote or rephrase any duplicated text outside the Methods section. Further consideration is dependent on these concerns being addressed.

We apologise for the overlapping text and have revised the relevant phrases. 

We confirm that there are no legal or ethical restrictions on sharing our data publicly. We have uploaded the data in fully anonymised form. 

Thank you. 

Reviewer #1: 

I would like to congratulate the authors on a very well written manuscript. It is well known in the literature that Acute Kidney Injury is prevalent in 5-20% of patients admitted to the ICU and this is generalized, not specific to cancer patients. This is a very difficult subset of patients to analyze in order to determine cause of mortality as multiple cofounding factors exist. The authors did a nice job of addressing the main limitations with the study but have a few comments/questions that the authors might address in the Discussion section.

1. The use of SCr only as a marker of renal function. The authors explained why they did not want to place foley catheters in many patients but as many cancer patients have cachexia and very small muscle mass which may affect SCr, knowing that the overall incidence of AKI could have been underestimated the correlation between AKI and mortality may have been overestimated. Could the authors comment on this.

The reviewer makes a very valid point. We have acknowledged in the limitations that we may have underestimated the incidence of AKI in this patient population. (page 15)

2. The inability to determine exact cause of death was also an issue as the majortiy of patients with AKI had Sepsis. Sepsis is known to be the most common cause of AKI in the ICU and often is a reversible cause of AKI. Were the authors able to determine if the sepsis was fully treated prior to death.

We thank the reviewer for this comment. 309 of 429 patients (72%) has sepsis on admission to ICU of whom 83 patients died in ICU. 75 (90%) patients had multi-organ failure at time of death. We have added this information. (page 7)

I am afraid we are not able to say whether sepsis was fully treated prior to death or still present and contributing to mortality.

3. There is significant data to support the use of early RRT once AKI is diagnosed. The authors mentioned both Intermittent modes as well as Continuous modes were utilized. Did they notice a difference in timing of initiation or mode of dialysis with respect to cause of or timing of mortality.

72 patients received RRT of whom 43 (59.7%) were started on continuous RRT within 24 hours of admission. Their ICU mortality was 44.2%. The remaining 29 patients were initiated on CRRT at a median time of 4 days. Their ICU mortality was 72.4% (p<0.05). We have added this information. (page 10)

Reviewer #2: 

The authors studied the impact of AKI in cancer patients admitted to an ICU and 28 day survival and overall survival. It has included both haematological malignancy (HM) and solid tumour (ST) for a total of 429 patients of which 60% has experienced AKI in total. Multivariable analysis showed that SOFA Score minus Renal on admission to ICU and AKI was an independent risk factor for both ICU and 28-day mortality. Interestingly, new AKI after 24 hours in ICU was associated with higher mortality than AKI on admission.

Comments: 

1. It has been demonstrated that AKI in cancer patients is associated with increased mortality. I agree this is one of few papers looking in to ICU admitted patients. The non-survivors overall and 28 days non-survivors had a higher median SOFA scores and AKI percentage. Interestingly the percentage of patients on RRT was higher in the non-survivors in both categories.

In the univariate analysis there is no comment about the use of RRT in this population and if it had a negative impact on survival would be interesting.

Initially, the authors do separate the hematological malignancies from solid tumor but in the analysis place them together. Typically patients with hematological malignancies are typically sicker and have poorer mortality in the ICU especially stem cell transplant patients; therefore, placing them all in the same category for survival analysis maybe inaccurate. I do agree that separating the in to hematological vs solid malignancy will limit numbers etc for the analysis.

We have repeated the multivariable analyses for hematological and solid tumour patients separately and added the results (Table 4).

2. Looking in to new AKI in the ICU is not well studied and this finding further contributes to the literature.

We thank the reviewer for this comment.

---

## [Decision Letter · Decision Letter 1]

14 Apr 2020

Acute kidney injury in critically ill cancer patients is associated with mortality: a retrospective analysis

PONE-D-20-01764R1

Dear Dr. Ostermann,

We are pleased to inform you that your manuscript has been judged scientifically suitable for publication and will be formally accepted for publication once it complies with all outstanding technical requirements.

With kind regards,

Emmanuel A Burdmann

Section Editor

PLOS ONE

Additional Editor Comments (optional):

Reviewers' comments:

Reviewer's Responses to Questions

**Comments to the Author**

1. If the authors have adequately addressed your comments raised in a previous round of review and you feel that this manuscript is now acceptable for publication, you may indicate that here to bypass the “Comments to the Author” section, enter your conflict of interest statement in the “Confidential to Editor” section, and submit your "Accept" recommendation.

Reviewer #1: All comments have been addressed

2. Is the manuscript technically sound, and do the data support the conclusions?

Reviewer #1: Yes

3. Has the statistical analysis been performed appropriately and rigorously? 

Reviewer #1: Yes

4. Have the authors made all data underlying the findings in their manuscript fully available?

Reviewer #1: Yes

5. Is the manuscript presented in an intelligible fashion and written in standard English?

Reviewer #1: Yes

6. Review Comments to the Author

Reviewer #1: The authors have adequately addressed all questions and comments to the reviewers. I would like to congratulate them on their work.

7. PLOS authors have the option to publish the peer review history of their article (what does this mean?). If published, this will include your full peer review and any attached files.

Reviewer #1: No

---

## [Editor Report · Acceptance letter]

11 May 2020

PONE-D-20-01764R1 

Acute kidney injury in critically ill cancer patients is associated with mortality: a retrospective analysis 

Dear Dr. Ostermann:

I am pleased to inform you that your manuscript has been deemed suitable for publication in PLOS ONE. Congratulations! Your manuscript is now with our production department. 

With kind regards,

on behalf of

Dr. Emmanuel A Burdmann 

Section Editor

PLOS ONE